# Toxicity Mechanisms of Gadolinium and Gadolinium-Based Contrast Agents—A Review

**DOI:** 10.3390/ijms25074071

**Published:** 2024-04-06

**Authors:** Susana Coimbra, Susana Rocha, Nícia Reis Sousa, Cristina Catarino, Luís Belo, Elsa Bronze-da-Rocha, Maria João Valente, Alice Santos-Silva

**Affiliations:** 11H-TOXRUN—1H-Toxicology Research Unit, University Institute of Health Sciences, Cooperativa de Ensino Superior Politécnico e Universitário (CESPU), Advanced Polytechnic and University Cooperative, CRL, 4585-116 Gandra, Portugal; 2Associate Laboratory i4HB—Institute for Health and Bioeconomy, Department of Biological Sciences, Faculdade de Farmácia da Universidade do Porto, 4050-313 Porto, Portugal; 3UCIBIO—Applied Molecular Biosciences Unit, Department of Biological Sciences, Faculdade de Farmácia da Universidade do Porto, 4050-313 Porto, Portugal; 4Departamento de Ciências e Tecnologia da Saúde, Instituto Superior Politécnico de Benguela, Benguela, Angola; 5National Food Institute, Technical University of Denmark, Kongens Lyngby, 2800 Copenhagen, Denmark

**Keywords:** gadolinium, toxicity mechanisms, gadolinium-based contrast agents, nephrotoxicity, magnetic resonance imaging

## Abstract

Gadolinium-based contrast agents (GBCAs) have been used for more than 30 years to improve magnetic resonance imaging, a crucial tool for medical diagnosis and treatment monitoring across multiple clinical settings. Studies have shown that exposure to GBCAs is associated with gadolinium release and tissue deposition that may cause short- and long-term toxicity in several organs, including the kidney, the main excretion organ of most GBCAs. Considering the increasing prevalence of chronic kidney disease worldwide and that most of the complications following GBCA exposure are associated with renal dysfunction, the mechanisms underlying GBCA toxicity, especially renal toxicity, are particularly important. A better understanding of the gadolinium mechanisms of toxicity may contribute to clarify the safety and/or potential risks associated with the use of GBCAs. In this work, a review of the recent literature concerning gadolinium and GBCA mechanisms of toxicity was performed.

## 1. Introduction

The paramagnetic properties of gadolinium (Gd (III)) has made it a crucial imaging aid tool for medical diagnosis and for treatment monitoring, across multiple clinical settings.

In the 1960s, the toxicity of Gd (III) in the salt form was reported in animal studies, limiting its use [1]. The development of the first formulations of Gd (III) stabilized by chelating agents, in the late 1980s, renewed the applicability of Gd (III) as a contrast agent [1]. 

Contrast agents composed of chelated Gd (III), commonly referred to gadolinium-based contrast agents (GBCAs), have been widely used in magnetic resonance imaging (MRI) for over three decades, supporting the diagnosis of tumors, central nervous system diseases, vascular diseases, bone marrow disorders, sclerosis, and cerebrovascular events [2], among other clinical conditions. Furthermore, recently, its potential applicability in the theranostic agents field has also been investigated [3,4]. The design of GBCAs must consider that the release of Gd (III) from chelates should be low enough to be safe; thus, Gd (III) must attach firmly to a high affinity ligand to form a safe GBCA. By chelating Gd (III), its toxicity is reduced to a safe level, while maintaining the paramagnetic properties, which will increase the sensitivity and specificity of MRI diagnostic. 

Considering the chemical structure of the chelating molecule, GBCAs can be classified as linear or macrocyclic, depending on whether or not they have an open or an enclosing structure, respectively (Figure 1). Depending on their charge, they can be ionic, like the acidic GBCA, or non-ionic, like the chelating agents with amide or alcohol groups. Linear complexes are flexible open chains that do not bind robustly to Gd (III), while macrocyclic GBCAs, with pre-arranged rigid rings, present almost the ideal size to trap the ion, offering a stronger linkage to Gd (III). The development of macrocyclic chelates was prompted by the low stability of linear GBCAs. Indeed, Gd (III) dissociates more quickly and easily from linear chelates, leading to higher circulating levels and increased tissue uptake of free Gd (III), which may entail long-term disturbances in multiple organs [5]. Studies with fibroblasts and macrophages showed that, following endosomal internalization into living cells, acyclic GBCAs are degraded much more rapidly than macrocyclic chelates [6]. 

Accumulation of Gd (III) ion has been reported in kidney [7,8], brain [9], liver [10], skin [11], and bone tissue [12]. Animal studies have shown that the amounts of Gd (III) retained in the organs are higher for linear GBCAs than for those with a macrocyclic structure [13,14,15]. In postmortem studies of patients who died from nephrogenic systemic fibrosis (NSF), a clinical complication that can be observed in subjects with compromised renal function, after exposure(s) to GBCAs [7], Gd (III) was found in all analyzed tissues, showing very high levels in the kidney, heart, and blood vessels [16]. The long-term retention of Gd (III) raises concerns about the safety of GBCAs, once the mobilization of such deposits may result in adverse events, with variable onset. A study conducted in an aquatic environment suggested that the chelating structure of the contrast agent may affect cell growth, also raising some concerns about the safety of the ligand [17]. Most of the NSF reported cases were associated with the administration of non-ionic linear agents, such as gadodiamide and gadoversetamide, although some NSF cases have also been associated with gadopentetic acid, a linear ionic agent [18]; with macrocyclic GBCAs, there are less reported cases of NSF, and most of them are in gadolinium-exposed patients with renal insufficiency [19,20,21,22].

Reports of NSF occurrence in patients with advanced kidney disease exposed to GBCAs strengthened the concern on their nephrotoxicity [7,19,20,21,22]. A slower elimination of Gd (III), due to kidney dysfunction, increases the potential for Gd (III) accumulation in the kidney and other tissues [23]. Accumulation of Gd (III) in the kidney, as well as in other organs, has also been reported in individuals without renal dysfunction, particularly in those submitted to repeated administrations of GBCAs [24].

Understanding the pathways involved in the toxicity of Gd (III) might help to clarify the clinical significance of its renal retention, allowing a more accurate assessment of the risks associated with GBCAs use. This review aims to identify, gather, and summarize the current scientific data available on Gd (III) and/or GBCAs mechanisms of toxicity.

## 2. Gd (III) Mechanisms of Toxicity

To understand the cellular and molecular mechanisms of action underlying the toxic effects of Gd (III) and/or GBCAs, we performed a bibliographic search, considering in vitro and in vivo mechanistic studies, in the databases PubMed, Scopus, and Web of Science. Keywords were specifically used for each database in order to retrieve all studies containing information on Gd (III) or GBCA exposure. From this search, only non-human experimental mechanistic studies were included in this review.

Table 1 summarizes, chronologically and alphabetically (first author’s surname), the studies for Gd (III) mechanisms of toxicity deemed relevant for the purpose of this review.

A total of 93 studies were herein reviewed (Table 1), including studies concerning exposure to non-chelated Gd (III) (*n* = 54) and/or to GBCAs (*n* = 51). Sixty-four of these involved in vitro studies, using either established cell lines, primary cultures, or isolated tissues exposed to Gd (III) or GBCAs; two studies assessed the cellular mechanisms in hepatic material isolated from rats administered with Gd (III); 22 studies were conducted in vivo, using different species of animals; and five studies included both in vitro and in vivo models. Of note, in 15 of the in vivo studies, the effect of repeated administrations was evaluated, and animal models of renal failure were used in 6 studies.

According to the gathered data, several signaling pathways have been implicated in Gd (III) mechanisms of toxicity, such as MAPK/ERK (mitogen-activated protein kinase/extracellular signal-regulated kinase), PI3K/Akt (phosphoinositide-3-kinase/protein kinase B), and EGFR (epidermal growth factor receptor) signaling [26,54,55,71,84,87,89,90,93], suggesting that Gd (III) interferes with the transduction of molecules involved in the regulation of inflammatory processes, and in cell metabolism, proliferation, growth, and survival.

Upregulation of inflammation, oxidative stress, and apoptosis were highlighted as potential mechanisms of Gd (III) cytotoxicity [25,30,34,38,39,40,42,60,61,62,64,66,70,76,84,86,91,99,101,102,111,113]. It has been reported that exposure to Gd (III) or GBCAs may induce the expression of several profibrotic chemokines and cytokines, and alter cell growth [11,41,54,59,68,69,72,77,88,89,95,96,98,112], initiating and supporting tissue fibrosis, namely renal fibrosis [50,52], as occurs in NSF. These compounds are also capable of increasing the proliferation and activity of fibroblasts [65,74,82,97], favoring collagen production [72,82,92], of triggering skin fibrosis [53,57], and inducing the upregulation of biomarkers of fibrosis and inflammation, as observed in exposed macrophages and fibroblasts [57]. Other inflammatory changes have been highlighted, including the alterations in macrophage profile, though the effect on macrophage polarization, into M1 phenotype or anti-inflammatory M2 phenotype, is controversial [27,43,69]. Increase in liver M2 cells in aged animals [43], involvement of lysosomes in Gd (III) accumulation in macrophages and in their proliferation [100], leukocytic infiltration, at the renal level with tubules atrophy [50], enhancement of neutrophil elastase activity [35], and alteration in leukocyte count [37,68] were also reported. Exposure to Gd (III) prompted its phagocytosis by the mononuclear phagocytic system [110]. According to Wang et al. [56], ongoing inflammation seems to facilitate the retention of Gd (III) in the brain tissue.

GBCAs and Gd (III) were seen to promote the production of reactive oxygen species (ROS), nitrate/nitrite, and prostaglandin E2; increase thiobarbituric acid reactive substances (TBARS) levels; and inhibit nitric oxide formation [42,61,62,75,91,107]. Increased levels of ROS were identified as the initiating event of Gd (III)-induced apoptosis [101]. Besides lipid peroxidation and ROS production, Gd (III) prompted the formation of autophagic vesicles, also revealing apoptotic and necrotic potential [42,110], pointing towards a multitude of cell death pathways being activated. Indeed, a decrease in cell viability, an increase in cell death through apoptosis, and autophagic activation have been associated with Gd (III) toxicity [10,30,34,40,46,47,48,51,58,60,64,70,99,113]. Mitochondrial dysfunction [38,42,51,91,108,109] and suppressing mitochondria membrane potential [62,101] were also described. Besides cytotoxic, genotoxic potential has also been attributed to Gd (III) exposure [28,45,62], and DNA cleavage of peripheral blood lymphocytes was reported [70,103].

Gd (III) was shown to interfere with calcium homeostasis as well: competition of Gd (III) with calcium, needed for cellular processes, was highlighted as a potential mechanism of cytotoxicity [49,58,89,105,114,115]. Promotion of calcium influx was also reported [79], along with inhibition of mitochondrial calcium-activated F_1_F_O_-ATPase and desensitization of the permeability transition pore to calcium by binding to F_1_ [32], which is also in line with the reported mitochondrial dysfunction. Gd (III) may block calcium transport in tissues with a lower excretion rate, increasing toxicity; it may inhibit some enzymes that are activated by calcium, interfering with the reticuloendothelial system, as well as with other calcium-dependent biological processes [33,36]. It can, also, disturb physiological processes, like contraction of smooth, skeletal, and cardiac muscles; transmission of nervous influx; and blood coagulation [116]. Furthermore, cell culture studies have shown that Gd (III) may lead to abnormal calcification of several types of cultured cells, inducing calcium deposition [83]. NSF may be, at least in part, a consequence of this alteration in the calcification process, which promotes hardening of the skin and fibrotic changes in other tissues and organs.

Other plausible mechanisms underlying Gd (III) toxicity include blockage of adenosine diphosphate and adenosine triphosphate (ATP) hydrolysis through stimulation of angiotensin II AT1 receptors [78]; inhibition of ATP-permeable channels [106]; interference with the epithelial Na^+^-channel’s activity [63]; downregulation of RhoA, mTORC1, and mTORC2 proteins [31]; and inhibition on both inward and outward ionic current through Gd (III) accumulation at the binding site of the Na^+^-Ca^2+^ exchanger protein that carries the current [104]. It may also interfere with the mobilization of iron [29,67], as it has been associated with total iron-binding capacity (TIBC) decrease [75], and to increases in serum iron, ferritin [41,75,80], and transferrin saturation [75]. It also has an effect on the differentiation of mononuclear cells into ferroportin-expressing fibrocytic cells [67] and the differentiation of mononuclear cells into collagen-secreting cells, with increased expression of iron metabolism proteins and of angiogenic and osteoblast-lineage markers [73]. Iron involvement in Gd (III) toxicity is in line with the transmetallation theory, by which endogenous metals, like iron, zinc, copper, magnesium, or calcium, attract the ligand, replacing gadolinium with the release of free Gd (III), which may deposit in different tissues. Finally, metabolic dysfunction, affecting lipid metabolism [38,52], and glycolytic and redox metabolic pathways were also highlighted [81]. Tubulin was pointed to as a potential Gd-binging protein, at least in the NIH-3T3 cells (mouse embryonic fibroblasts); this binding might inhibit the assembling of tubulin or depolymerize microtubules in cells [44].

The pathophysiology of NSF remains poorly clarified and appears to be independent of sex, race, or age [9]. The dissociation of Gd (III) from Gd-chelates, which has been highlighted as the primary etiology, is more likely to occur in patients with renal dysfunction, who have a reduced excretion rate, allowing a longer retention that facilitates in vivo ion dissociation, when compared to those with normal renal function.

Few studies using animal models of renal failure have addressed the impact of kidney disease in the toxicity of GBCAs [11,35,57,67,75,76]. Nonetheless, they suggested that, in the case of renal disease, GBCAs decreased renal function [75], triggered skin fibrosis [11,57], increased the number of fibrocytes (related to the oxidative stress environment) [76], enhanced the differentiation of mononuclear cells into ferroportin-expressing fibrocystic cells [67], produced renal tube vacuolization [11], and caused disturbances in iron metabolism and TBARS values [75], as well as increased neutrophil count and neutrophil elastase activity [35].

Accumulation of Gd (III) in the kidney, as well as in other organs, has also been described in individuals without renal dysfunction [24], particularly in individuals submitted to repeated administrations of GBCAs [24]. According to Roberts et al., in subjects with normal renal function, exposure to large cumulative doses of GBCAs can lead to the deposition of Gd (III) in the skin and brain [117]. The observation of renal damage and tissue accumulation of Gd (III) after GBCA exposure, in subjects without previous renal disease, suggests the involvement of other nephrotoxic mechanisms, beyond the decrease in Gd (III) elimination, due to impaired renal function.

Cell culture studies using Gd (III)-exposed HK-2 cells (human proximal tubular cell line) reported increased oxidative stress, mitochondrial dysfunction, cell death by apoptosis, switching to necrosis at higher Gd (III) levels, and autophagic activation. Disturbance of the lipid metabolism was also observed, with intracellular accumulation of lipid droplets and upregulation of genes related to both lipogenesis and lipolysis; moreover, increased expression of the modulators of various signaling pathways involved in the development and progression of renal disease, including inflammation, hypoxia, and fibrosis, were also detected, even at subtoxic concentrations [38].

## 3. Concerns about the Use of GBCAs

The ability of Gd (III) to be retained in body tissues following its detachment from linear GBCAs led the European Medicines Agency (EMA) to recommend a restriction in their use [118]. Some linear structure contrast agents, namely gadodiamide and gadoversetamide, were suspended. According to the EMA, the use of gadoxetic and gadobenic acid should be restricted to liver MRIs, as they undergo biliary excretion, meeting an important diagnostic need; gadopentetic acid should be restricted to intra-articular administration for MRI of the joints, since the dose necessary for this exam is very low. Moreover, the EMA recommended the use of agents with a macrocyclic molecular structure (such as gadoteric acid, gadobutrol, and gadoteridol), at the lowest dose necessary for diagnosis, and only if this is not possible without resorting to contrast agents.

Although no restrictions were made for the use of macrocyclic GBCAs, a few human and animal studies have already demonstrated that their use leads to Gd (III) retention in body tissues [119], which was also reported in patients with normal renal function [120]. Following the administration of macrocyclic GBCAs in rats, organ tissue (e.g., brain and renal, hepatic, and splenic tissues) deposition was observed [121]. In adults and children, after multiple administrations of macrocyclic GBCAs, an increase in signal intensity on unenhanced T1-weighted magnetic resonance in the brain was detected [122,123,124,125], suggesting Gd (III) retention at this organ. Ex vivo analysis of brain and bone tissues from patients administered with the macrocyclic GBCAs, gadobutrol or gadoteridol, showed Gd (III) deposition [120]. The development of NSF, following the use of macrocyclic GBCAs, has also been reported, although data is not always consensual [126].

The mechanistic studies presented in Table 1 suggest that some macrocyclic GBCAs appear to be more stable, as expected, with lower propensity to release Gd (III) [37], and with safer profiles when compared to linear GBCAs [33,37,60,61,63,77,82,85].

Nevertheless, it was reported that both linear and macrocyclic GBCAs stimulated the expression of multiple type I interferon-regulated genes and of numerous chemokines, cytokines, and growth factors in normal human blood monocytes [98]; in addition, supernatants recovered from monocyte cell cultures exposed to both types of GBCAs stimulated the expression, in normal dermal fibroblasts, of types I and III collagen, fibronectin, and α-smooth muscle actin [72]. Although gadodiamide, a linear GBCA, led to greater skin fibrosis and dermal cellularity than the macrocyclic gadoteridol, both led to renal proximal tubule vacuolization and increased fibronectin accumulation [11]. In addition, kidneys showed a significantly higher Gd (III) content after administration of gadodiamide and of the macrocyclic gadobutrol, as compared to gadobenate dimeglumine administration [127].

Repeated exposure to the macrocyclic GBCAs gadoteric acid or gadobutrol caused elevation of oxidative stress and inflammation in the brain [39]; although neurotoxicity was more prominent for linear GBCAs, both linear and macrocyclic GBCAs triggered neuronal cell death, through activation of apoptosis [34].

In rat testis, gadoteric acid and gadodiamide induced apoptosis in the Leydig cells, increased serum calcium levels, and reduced testosterone levels [58]. These GBCAs were also able to trigger hepatocellular necrosis and apoptosis, causing liver damage [10].

It should be taken in consideration that different profiles have been reported for macrocyclic GBCAs; for instance, gadobutrol appears to easily release Gd (III), while gadoterate meglumine showed the best performance concerning the complex stability [49]; gadoteric acid neurotoxic potential was found to be higher than that of gadobutrol [39]. It is important to highlight that each GBCA has its own properties and its own behavior regarding in vivo retention or deposition. The results reported for one GBCA cannot be extrapolated for all GBCAs [128]. For instance, gadoteric acid undergoes a much faster residual excretion from the body than linear GBCAs [129]. Gadoteridol was found to be eliminated more rapidly from rat cerebellum, cerebrum, and skin, compared to gadoteric acid and to gadobutrol, in the first 5 weeks after repeated administration of these macrocyclic GBCAs, resulting in lower levels of retained Gd (III) in these tissues [130]. The faster clearance of gadoteridol has been attributed to its lower viscosity, molecular weight, and osmolality [131]. Even among macrocyclic GBCAs, there are differences in their clearance and in the amount of Gd (III) retention [130].

The concerns regarding GBCA safety are driving research to find other solutions, with better safety and pharmacokinetic profiles, improving their performance and/or reducing the administered dose; these novel agents may include macro- and supramolecular multimeric Gd (III) complexes (dendrimers, polymers, carbon nanostructures, micelles, and liposomes) [132]. The encapsulation of Gd (III) into nanoparticulates is another approach considered to overcome the poor selective tissue labeling and localization associated with GBCAs [133]. Smart radiotherapy biomaterials loaded with Gd-based nanoparticles were also investigated for use in MRI scans, revealing a great potential [134]. The interest of liposomal formulations application in MRI has been increasing [135]. For instance, the use of nanoliposomal Gd (III) did not present adverse effects on human-derived hepatocyte-like HepaRG cells and macrophages, although in vitro studies are needed to evaluate its safety [136].

Another approach to reduce GBCAs toxicity is the association with antioxidant or metal chelation agents. A study, in rats with renal failure (5/6 nephrectomy), exposed to a single dose of gadoteric acid, showed that the effects in renal function improved by treatment with the antioxidant N-acetylcysteine [75]. Renal failure (5/6 nephrectomy) mice exposed repeatedly to gadodiamide developed NSF, infiltration of ferroportin-expressing fibrocyte-like cells, and iron accumulation in the skin; these effects were less pronounced in the group treated with gadodiamide plus deferiprone [67], a metal chelating agent used in clinical practice to treat iron overload, able to avoid Gd (III) tissue deposition. In accordance, the addition to gadodiamide of the chelating agent DTPA (diethylenetriaminepentaacetic acid) reduced transmetallation of this GBCA [137], suggesting that the use of metal chelates may help to reduce, possibly even eliminate, Gd (III) retention by tissues.

## 4. Final Considerations

Considering the usefulness of contrast agents, the lack of safer alternatives to GBCAs and the higher prevalence of renal complication in GBCA-exposed patients, especially in patients with renal insufficiency when the incidence of chronic kidney disease is increasing worldwide, the studies on the molecular and cellular mechanisms underlying Gd (III) cytotoxicity for each GBCA, as well as their pharmacological effects, are warranted.

This review provides an overview of the available evidence regarding the toxicity mechanisms of Gd (III) and GBCAs determined using in vitro and in vivo models, providing scientific grounds for the development of counteracting therapeutic measures.

It is clear that, compared to GBCAs with macrocyclic structures, the linear GBCAs are more unstable and, thereby, have shown higher Gd (III) retention and cytotoxicity in the organs.

Cell cultures with macrophages and renal and endothelial cells demonstrate that GBCA toxicity seems to involve pro-inflammatory and pro-fibrotic mechanisms. Despite several studies involving cell cultures, fewer have tackled in vivo evaluation using animal models, particularly addressing renal function. Current available data indicate that single exposure to macrocyclic GBCAs seems safe in animals with normal renal function. However, the toxicity at long-term Gd (III) retention deserves more investigation, both in cases with normal and decreased renal function.

In patients with moderate/severe renal disease, GBCA exposure may further compromise renal function, but the effect in preexisting mild kidney disease is not so clear. Although there are studies reporting nephrotoxicity and impaired renal function associated with repeated administrations of GBCAs, the frequency of exposure used in most research studies poorly mimics the use of these agents in clinical practice, and some research studies were carried out in models of advanced stage of renal disease. Also, the use of different lengths of exposure to GBCAs makes the interpretation and comparison between studies difficult. The effect of repeated administrations in mild kidney disease using standardized exposures to contrast agents deserves further study. Finally, considering the increasing prevalence of chronic kidney disease worldwide and that most of the complications following GBCA exposure are associated with renal dysfunction, the mechanisms underlying GBCA toxicity, especially renal toxicity, need further research studies.

## Figures and Tables

**Figure 1 ijms-25-04071-f001:**
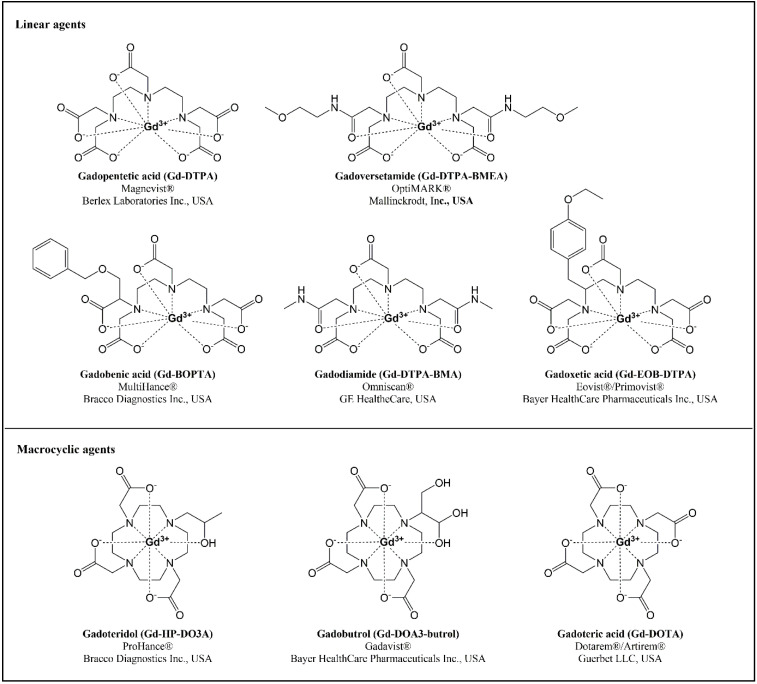
Chemical structures of linear and macrocyclic gadolinium-based contrast agents, their brand names, and registering pharmaceutical companies.

**Table 1 ijms-25-04071-t001:** In vitro and in vivo studies concerning gadolinium mechanisms of toxicity.

Reference	Study Design	Main Findings
Akhtar et al., 2022 [25]	Human monocytes (THP-1 cell line) exposed to nanoparticles (NPs) of CeO_2_ or Gd_2_O_3_	Gd_2_O_3_ NPs showed increased cytotoxic, pro-inflammatory (↑IL-1β and TNFα), and oxidative (↑ROS and TBARS, ↓GSH) potential, compared to CeO_2_ NPs; cell death induced by Gd_2_O_3_ NPs appears as apoptosis-independent (no effect on Bax-Bcl2 or caspase 3 activity), contrarily to CeO_2_ NPs
Ariyani et al., 2022 [26]	Rat glioma cells (C6 cell line), human astrocytoma cells (U87MG cell line), and primary cultures of mouse cerebral cortex astrocytes, exposed to Omniscan™ (gadodiamide), Magnescope^®^ (gadoteric acid), Magnevist^®^ (gadopentetic acid), or Gadovist^®^ (gadobutrol)	All GBCAs acted via integrin αvβ3, leading to increased astrocytes migration, focal adhesion, and F-actin rearrangement, through activation of FAK/ERK1/2/Akt and Rho family of GTPases signaling pathways
Chanana et al., 2022 [27]	Mouse peritoneal macrophages isolated from C57BL/6 (H-2b) mice and murine leukemia transformed mouse macrophages (RAW 264.7 cell line) exposed to Dotarem^®^ (gadoteric acid) in the presence of a static magnetic field gradient	Gadoteric acid appeared to affect actin polymerization, leading to macrophage elongation and relocation of organelles; enhanced pro-inflammatory M1 phenotype (↑iNOS and CD80) and decreased anti-inflammatory M2 phenotype (↓FcεRI); the magnetic field gradient had an opposite effect
Cobanoglu 2022 [28]	Human peripheral blood lymphocytes exposed to Dotarem^®^ (gadoteric acid) and OptiMARK^®^ (gadoversetamide)	Gadoversetamide, but not gadoteric acid, showed genotoxic and cytotoxic potential (↑frequency of micronuclei, nucleoplasmic bridges and nuclear buds, ↓cytostasis)
Nakamura et al., 2022 [29]	BALB/c male mice treated with a single administration of Omniscan™ (gadodiamide), Gadovist^®^ (gadobutrol), or Gd (III) in the form of Gd(NO_3_)_3_ or GdCl_3_	Tissue deposition of gadolinium varied with the chemical forms tested—higher levels for Gd(NO_3_)_3_, spleen enlargement and iron deposition for Gd (III)-treated mice
Tsai et al., 2022 [30]	Human keratinocytes (HaCaT cell line) exposed to gadodiamide	Apoptotic cell death (↑caspase 3 activity, ↓Bcl-2, ↑Bax) and autophagic activation (↑autophagic vacuoles and acidic lysosomes); autophagy potentiated apoptotic cell death
Uosef et al., 2022 [31]	Mouse macrophages treated with Dotarem^®^ (gadoteric acid)	Macrophages retained Gd (III) for at least 7 days after exposure; this retention downregulated the expression of RhoA, mTORC1, and mTORC2 proteins, and dysregulated the expression level of organelle markers
Algieri et al., 2021 [32]	Mitochondrial (MT) fractions from swine hearts (*Susscrofa domesticus*) exposed to GdCl_3_	GdCl_3_ inhibited both MT Ca^2+^- and Mg^2+^-activated F_1_F_O_-ATPase and desensitized the permeability transition pore to Ca^2+^ by binding to F_1_
Baykara et al., 2021 [33]	Mouse hypothalamic neurons (GT1-7 cell line) treated with Omniscan™ (gadodiamide) or Dotarem^®^ (gadoteric acid)	The amount of gadolinium released from gadodiamide was higher (versus gadoteric acid), leading to a higher impact in Ca^2+^ signaling
Erdoğan et al., 2021 [34]	Human neuroblastoma cells (SH-SY5Y cell line) exposed to Dotarem^®^ (gadoteric acid), Gadovist^®^ (gadobutrol), Omniscan™ (gadodiamide), Primovist^®^ (gadoxectic acid), Magnevist^®^ (gadopentetic acid), or OptiMARK™ (gadoversetamide)	Both linear and macrocyclic GBCAs triggered neuronal cell death through activation of apoptosis (↑Bax/Bcl-2 ratio); neurotoxicity was more prominent in cells exposed to linear GBCAs
Kartamihardja et al., 2021 [35]	Renal failure mouse model (kidney electrocoagulation) exposed for three weeks to Omniscan™ (gadodiamide) and Magnevist^®^ (gadopentetic acid), three times per week	Gadodiamide showed higher skin gadolinium retention than gadopentetic acid, and more prominent pro-fibrotic potential (↑Collagen 1α, CTGF, TGFβ, αSMA, and IL-6); both GBCAs, especially gadodiamide, increased skin infiltration of CD3+ T cells and CD68+ macrophages, and (skin) expression and (serum) activity of neutrophil elastase
Kartamihardja et al., 2021 [36]	Primary mouse pups’ cerebellar cultures exposed to Magnevist^®^ (gadopentetic acid) or Gadovist^®^ (gadobutrol), in the presence or absence of iron (II)	Both GBCAs augmented dendrite arborization; iron (II) potentiated this effect only with gadopentetic acid
Kong et al., 2021 [37]	ICR female mice treated with repeated administrations of Magnevist^®^ (gadopentetic acid), Dotarem^®^ (gadoteric acid), Omniscan™ (gadodiamide), or Gadavist^®^ (gadobutrol) for 3–5 weeks, followed by a recovery period of 1–5 weeks	Gadodiamide caused vacuolar changes in renal tubular epithelium; linear GBCAs increased leukocyte count after 5 weeks of exposure and induced higher gadolinium tissue deposition (cerebellum, liver, kidney, femur, skin, and peripheral nerve) compared to macrocyclic GBCAs
Reis Sousa et al., 2021 [38]	Human proximal tubular cells (HK-2 cell line) exposed to GdCl_3_	GdCl_3_ induced disruption of oxidative status (↓TAS and GSH, ↑GSSG and NRF2), MT dysfunction (↑Ca^2+^, ↓ΔΨ_m_ and ATP), cell death by apoptosis (↑caspase 3, ↓Bcl-2), switching to necrosis (↑LDH leakage) at higher levels, and autophagic activation (↑p62); disturbance of lipid metabolism (↑ACACA, CPT1A, and neutral red uptake) increased expression of modulators of inflammation, hypoxia, and fibrosis (↑NFκB, IL-6 and 1β, TGFβ, OPN, and HIF-1α) at low to subtoxic concentrations
Solmaz et al., 2021 [39]	Male Sprague Dawley rats treated repeatedly for 3 weeks with Gadovist^®^ (gadobutrol), Clariscan^®^ (gadoteric acid), and Dotarem^®^ (gadoteric acid); evaluation after a recovery period of 1 week	Repeated exposure to GBCAs caused hippocampal gliosis and increased oxidative stress and inflammation in the brain (↑LPO and TNFα, ↓SOD activity); neurotoxicity of gadobutrol was relatively lower than that of gadoteric acid
Tsai et al., 2021 [40]	Human fetal normal glial cells (SVG P12 cell line) exposed to Omniscan™ (gadodiamide)	Apoptotic cell death (↓Bcl-2 and -X_L_, ↑Bax and BAD, ↑cytochrome *c*, Apaf-1, and cleaved-caspase 3 and 9) and autophagic activation (↑autophagic vacuoles and acid lysosomes, ↑LC3-I/II turnover, beclin-1, autophagy-related proteins -5, and -14); autophagy potentiated cell death
Xie et al., 2021 [41]	Healthy mice treated with repeated doses of γ-Fe_2_O_3_ NPs and gadopentetic acid (Gd-DTPA)	Proinflammatory responses elicited by Gd-DTPA were stronger than for γ-Fe_2_O_3_ NPs (↑IL-1β, -6, -18, TNFα, CRP, and ferritin)
Akhtar et al., 2020 [42]	Human umbilical vein endothelial cells (HUVEC cell line) exposed to Gd_2_O_3_ NPs	Gd_2_O_3_ NPs acted as inducer of oxidative stress (↑TBARS, ROS and LPO, ↓GSH), MT dysfunction (↑MT membrane potential), and autophagy (↑acidic lysosomes and autophagic vacuoles), and revealed apoptotic (↑caspase 3 and annexinV) and necrotic potentials
Bloomer et al., 2020 [43]	Hepatic macrophages of young (6 months) and aged (24 months) Fischer 344 rats evaluated 2 days after exposure to GdCl_3_	In aged animals, GdCl_3_ shifted liver macrophage polarization towards the anti-inflammatory M2 phenotype (↓iNOS^+^ cells).
Nong et al., 2020 [44]	Mouse embryo fibroblasts (NIH-3T3 cell line) exposed to gadodiamide or GdCl_3_	Inhibition of cell growth, more pronounced with GdCl_3_; tubulin filaments appeared as potential gadolinium-binding proteins, which might lead to impaired microtubule assembling
Siew et al., 2020 [45]	Chinese hamster lung fibroblasts (V79-4 cell line) exposed to GdCl_3_	Cell death and no significant DNA damage, although showing clastogenic potential (↑micronuclei frequency)
Supawat et al., 2020 [46]	K562 cancer cells and red blood cells exposed to gadoteric acid, gadopentetic acid, or gadobenic acid	Gadoteric acid and gadobenic acid decreased cell viability in K562 cancer cells in a concentration-dependent manner
Takanezawa et al., 2020 [47]	Human embryonic kidney cells (HEK293 cell line), lung carcinoma epithelial cells (A549 cell line), neuroblastoma cells (SH-SY5Y cell line), and mouse embryonic fibroblasts (MEF cell line) exposed to Gd(NO_3_)_3_ or GdCl_3_	Gd (III) reduced cell viability in all cell lines, triggered ER stress, and activated autophagy (↑LC3-II), which appears as cytoprotective against Gd (III) toxicity
Akhtar et al., 2019 [48]	Human breast cancer cells (MCF-7 cell line) exposed to Gd_2_O_3_ NPs or to GdCl_3_	Gd_2_O_3_ NPs and GdCl_3_ induced cytotoxicity (↑LDH leakage), oxidative damage (↑TBARS, ROS, GSH), and autophagic activation (↑autophagic vacuoles and acidic lysosomes); cell death was apoptosis-dependent (↑Bax/Bcl2 ratio) for GdCl_3_ and apoptosis-independent for Gd_2_O_3_ NPs
Baykara et al., 2019 [49]	Primary cultures of dorsal root ganglion neuron exposed to gadolinium, Omniscan™ (gadodiamide), Dotarem^®^ (gadoteric acid), Gadovist^®^ (gadobutrol), or MultiHance^®^ (gadobenic acid)	Ca^2+^ levels within neurons decreased, as ionic currents were blocked by Gd (III) released from the chelates, in accordance with their stability (gadobutrol < gadobenic acid ≈ gadodiamide; no effect from gadoteric acid)
Beyazal Celiker et al., 2019 [50]	Male Sprague Dawley rats treated with repeated administrations of Dotarem^®^ (gadoteric acid) or Omniscan™ (gadodiamide) for 5 weeks; evaluation after a recovery period of 5 weeks	Gadodiamide promoted higher kidney interstitial fibrosis, amyloid deposits, and vasocongestion, while gadoteric acid led to greater renal leukocytic infiltration and tubules atrophy; both GBCAs increased caspase 3 expression
Bower et al., 2019 [51]	Differentiated human neuroblastoma cells (SH-SY5Y cell line) exposed to Omniscan™ (gadodiamide), Magnevist^®^ (gadopentetic acid), Primovist^®^ (gadoxetic acid), MultiHance^®^ (gadobenic acid), Dotarem^®^ (gadoteric acid), Gadovist^®^ (gadobutrol), or ProHance^®^ (gadoteridol)	GBCAs triggered cell death by apoptosis, with reduction of the ΔΨ_m_ and of the oxidative respiratory function; disturbances were dependent on the stability of the GBCA, being more pronounced for linear GBCAs
Do et al., 2019 [52]	Female C57 black mice exposed to repeated administrations of Omniscan™ (gadodiamide) for 4 weeks	Impaired renal function, associated with myeloid cell infiltration and renal fibrosis (↑fibronectin, CCR2, and αSMA); metabolic dysfunction was also induced, with particular impact on renal lipid metabolism; obesity appeared to amplify these effects
Do et al., 2019 [53]	Female C57 black mice exposed to repeated administrations of Omniscan™ (gadodiamide) for 8 weeks	Skin fibrosis mediated by CCR2 (↑fibronectin, collagen I, CCR2, CCL2)
Pan et al., 2019 [54]	Human embryonic kidney cells (HEK293 cell line) treated with GdCl_3_	Proliferation of HEK293 cells (increased DNA synthesis and activation of EGFR/Akt/ERK signaling pathways; pro-fibrotic/pro-inflammatory changes (↑TGFβ and its receptor, TNFα, TIMP-1, and integrins αV and β1))
Tsai et al., 2019 [55]	Rat glioma C6 cells treated with GdCl_3_	Cell death by apoptosis (↑caspases 3, 8, and 9 activity, ROS and Ca^2+^, ↓ΔΨ_m_); down-regulation of the mitogen-activated protein kinases pathway
Wang et al., 2019 [56]	SJL/J mice, healthy or with autoimmune encephalomyelitis, exposed to repeated administrations of gadopentetic acid for 4 days	Ongoing inflammation favored retention of Gd (III) in the brain tissue
Weng et al., 2019 [57]	Adenine-induced renal failure rat model treated with repeated administrations of gadodiamide for 5 days; human normal liver cells (L02 cell line), human embryonic kidney cells (HEK293 cell line), mouse fibroblasts (3T6 cell line), and mouse macrophages (RAW264.7 cell line), exposed to gadodiamide	Skin fibrosis, oxidative stress, and inflammation (↑αSMA and TGFβ1, heme oxygenase-1, NOX4, CCL2, IL-1β and TNFα) in renal failure rats; in vitro exposure of macrophages showed upregulation of markers of fibrosis and inflammation (↑αSMA and TGFβ1, IL-1β and TNFα), and of fibrosis (↑αSMA) in fibroblast exposed to the supernatant of exposed macrophages; at the highest concentrations, promoted cell death in normal liver and kidney cells and in macrophages
Beyazal Celiker et al., 2018 [58]	Male Sprague Dawley rats treated with repeated administrations of Dotarem^®^ (gadoteric acid) or Omniscan™ (gadodiamide) for 5 weeks	Both showed toxic effects on testis tissue, inducing apoptosis (↑caspase 3 and Ca^2+^) and reducing testosterone levels
Fattah et al., 2018 [59]	Human breast cancer (MCF-7 cell line), mammary epithelial (Hs 578T cell line), and epithelial-like triple-negative breast cancer cells (MDA-MB-231 cell line) exposed to gadopentetic acid	Triggered cell proliferation of MCF-7 cells at low concentrations and cell death, as well as cell migration, at higher levels
Friebe et al., 2018 [60]	Lymphocytes from healthy donors incubated with Gadovist^®^ (gadobutrol), Dotarem^®^ (gadoteric acid), Omniscan™ (gadodiamide), Magnograf^®^ (gadopentetic acid), or Primovist^®^ (gadoxetic acid), either alone or combined with ultra-high-field 7-T magnetic resonance imaging exposure	Only linear GBCAs showed a dose-dependent increase in apoptosis (↑annexinV^+^ cells) and a decrease in DNA synthesis, independent of additional 7-T magnetic resonance imaging co-exposure
Mercantepe et al., 2018 [10]	Male Sprague Dawley rats exposed repeatedly to Omniscan™ (gadodiamide) or Dotarem^®^ (gadoteric acid) for 20 days	Both triggered hepatocellular necrosis, portal inflammation, and apoptosis (↑caspase 3); no changes occurred in total antioxidant/oxidant capacity
Weng et al., 2018 [61]	Macrophages exposed to low levels of Omniscan^®^ (gadodiamide), Primovist^®^ (gadoxetic acid), Magnevist^®^ (gadopentetic acid), Gadovist^®^ (gadobutrol), or GdCl_3_	GdCl_3_ and GBCAs had no effect on cell viability, but promoted MT dysfunction and oxidative stress (↓ΔΨ_m_, and ↑ROS); GBCAs also triggered an inflammatory response (↑nitrate/nitrite, prostaglandin E2, IL-6, ↓IL-10)
Alarifi et al., 2017 [62]	Human neuroblastoma cells (SH-SY5Y cell line) exposed to Gd_2_O_3_ NPs	Cell death by apoptosis (↑caspase 3, ↓ΔΨ_m_ and Bcl2/Bax ratio), DNA damage, and oxidative stress (↑ROS, LPO, SOD and catalase, ↓GSH)
Knoepp et al., 2017 [63]	*Xenopus laevis* oocytes heterologously expressing human epithelial Na^+^-channels exposed to GdCl_3_, Magnevist^®^ (gadopentetic acid), Dotarem^®^ (gadoteric acid), or their chelates	GdCl_3_ triggered changes in epithelial Na^+^-channels-mediated currents and appeared to act on at least two binding sites; Gd (III) released from the linear GBCAs, but not from gadoteric acid, was sufficient to interfere with the channels’ activity
Nagy et al., 2017 [64]	Human skin keratinocytes (HaCaT cell line), human limbal stem cells (HuLi cell line), colorectal adenocarcinoma (CaCO2 cell line), murine squamous carcinoma (SCC cell line), and Indian muntjac cells (IM cell line) exposed to GdCl_3_	Loss of cellular motility, premature chromatin condensation, and highly condensed chromatin, consistent with apoptotic cell death
Ozawa et al., 2016 [65]	Normal human dermis-derived fibroblasts incubated with Omniscan™ (gadodiamide)	Increased fibroblast growth, with increased DNA synthesis
Tsai et al., 2016 [66]	Human osteosarcoma cells (U-2 OS cell line) exposed to GdCl_3_	Apoptotic cell death mediated by death receptors, mitochondria, and ER stress (↑caspases 3, 4, 8, and 9 activity, Fas and its ligand, cytochrome *c*, Apaf-1, GADD153, GRP78, Ca^2+^, ↓ΔΨ_m_)
Bose et al., 2015 [67]	Male BALB/c mice with a two-step surgical 5/6 nephrectomy, exposed to repeated administrations of Omniscan™ (gadodiamide), with or without deferiprone, for 22 days; evaluations after 16 weeks; human peripheral blood mononuclear cells exposed to Omniscan™ (gadodiamide), with or without deferiprone	Renal failure mice exposed to gadodiamide developed nephrogenic systemic fibrosis; infiltration of ferroportin-expressing fibrocyte-like cells and iron accumulation in the skin; these effects were less pronounced in gadodiamide plus deferiprone-treated group; gadodiamide also prompted release of catalytic iron in vitro
Chen et al., 2015 [68]	BALB/c mice exposed to a single dose of gadopentetic acid for 24 h	Reduced circulating leukocytes and triggered an inflammatory response (↑IL-6 and TNFα); it also induced damage in the lungs, kidneys, and spleen
Schmidt-Lauber et al., 2015 [69]	Bone marrow derived macrophages from C57BL/6, Nlrp3−/−, and Asc−/− mice incubated with Omniscan™ (gadodiamide), gadopentetic acid, or GdCl_3_; male C57BL/6 and Nlrp3−/− mice intraperitoneally injected with a single dose of gadopentetic acid	Free Gd (III) and GBCAs induced the secretion of IL-1β in wild type mice-derived macrophages, through the activation of the inflammasome; Gd-containing compounds exhibited higher potential to activate anti-inflammatory M2 macrophages; the inflammatory response in vivo was also dependent on engagement of the inflammasome
Cho et al., 2014 [70]	Human lymphocytes exposed to GdCl_3_	Genotoxicity (↑micronuclei frequency and DNA damage), apoptotic cell death, and oxidative stress (↑ROS); extremely low-frequency electromagnetic fields potentiated these effects
Do et al., 2014 [11]	Human foreskin fibroblasts incubated with Omniscan™ (gadodiamide) or ProHance^®^ (gadoteridol); Female Fisher 344 rats with renal failure (5/6 nephrectomy) exposed to repeated doses of the GBCA for 4 weeks	In vitro, GBCAs triggered fibrosis (↑fibronectin, TGFβ, and αSMA); in vivo, gadodiamide led to greater skin fibrosis (↑fibronectin) and dermal cellularity than gadoteridol; gadoteridol induced higher expression of skin TGFβ and fibronectin accumulation in the liver; both agents led to proximal renal tubule vacuolization
Shen et al., 2014 [71]	Mouse embryo fibroblasts (NIH3T3 cell line) exposed to GdCl_3_	Cell proliferation via Rac, PI3K/Akt, and integrin-mediated signaling pathways
Wermuth and Jimenez 2014 [72]	Human dermal fibroblasts incubated with supernatants of human peripheral blood mononuclear cells treated with gadopentetic acid, Omiscan™ (gadodiamide), Dotarem^®^ (gadoteric acid), MultiHance^®^ (gadobenic acid), ProHance^®^ (gadoteridol), OptiMARK^®^ (gadoversetamide), or non-chelated Gd (III)	GBCA exposure led to variable expressions of profibrotic and proinflammatory cytokines in monocytes, more pronounced for linear agents (↑IL-4, -6, -13, TGFβ, and VEGF); overall increase in gene expression of cytokines, chemokines, genes involved in the activation of NFκB and interferon-responsive genes was also observed in Gd-treated monocytes; fibroblast showed a profibrotic phenotype (↑types I and III collagen, fibronectin, and αSMA)
Swaminathan et al., 2013 [73]	Human peripheral blood mononuclear cells exposed to Omniscan™ (gadodiamide); skin biopsy specimens from NSF patients (for confirmatory purposes)	Differentiation of mononuclear cells into collagen-secreting cells, with increased expression of iron metabolism proteins, angiogenic and osteoblast-lineage markers; these types of cell were also present in skin biopsies of NSF patients
Bleavins et al., 2012 [74]	Human dermal fibroblasts and epidermal keratinocytes isolated from neonatal foreskin exposed to Gd (III) salts, Magnevist^®^ (gadopentetic acid), MultiHance^®^ (gadobenic acid), Omniscan™ (gadodiamide), or non-clinical gadodiamide	Gd (III) salts attached to fibroblasts surface; proliferation was stimulated at lower concentrations via MAPK and PI3K signaling pathways, while cytotoxicity was seen at higher levels; GBCAs, but not the salts, also showed proliferative potential in fibroblasts under low-Ca^2+^ conditions, more evident for gadodiamide; no effects were observed in keratinocytes
Pereira et al., 2012 [75]	Male Wistar rats without or with renal failure (5/6 nephrectomy), exposed to a single dose of Dotarem^®^ (gadoteric acid)	Rats with renal failure showed a decreased renal function (↓GFR, ↑proteinuria, decrease in total iron binding capacity, increased serum ferritin, transferrin oversaturation, and increased plasmatic TBARS); treatment with the antioxidant N-acetylcysteine ameliorated these effects; rats with normal renal function showed no effects when treated with gadoteric acid compared to controls
Wagner et al., 2012 [76]	Female Fischer 344 rats with renal failure (5/6 nephrectomy) treated with repeated administrations of gadodiamide for 4 weeks	Skin presenting bone marrow-derived cells, with increased expression of αSMA, and with profibrotic (↑fibronectin, collagen IV, cathepsin L), and pro-oxidant phenotypes (↑superoxide, NOX4)
Wermuth & Jimenez 2012 [77]	Human embryonic kidney cells (HEK293 cell line) expressing one of different human TLRs or NLRs, and macrophages differentiated from human peripheral blood mononuclear cells exposed to Dotarem^®^ (gadoteric acid), MultiHance^®^ (gadobenic acid), ProHance^®^ (gadoteridol), OptiMARK^®^ (gadoversetamide), Omniscan™ (gadodiamide), non-clinical gadodiamide or gadopentetic acid, or non-chelated Gd (III)	Non-chelated Gd (III), gadoteric and gadobenic acid, as well as both gadodiamide formulations, induced NFκB activation via TLR4 and 7, more pronounced with the latter two; this stimulation of TLR resulted in a strong profibrotic/pro-inflammatory response in macrophages treated with Omniscan™ and gadodiamide (↑CXCL10, 11, and 12, CCL2 8 and 9, IL-4 and -6, TGFβ, and VEGF)
Angeli et al., 2011 [78]	Aortic rings of Wistar rats incubated with GdCl_3_	Blockade of ADP and ATP hydrolysis through stimulation of angiotensin II receptor type 1
Feng et al., 2011 [79]	Primary cultures of cortical astrocytes, isolated from neonatal Sprague Dawley rats, treated with GdCl_3_	Ca^2+^ influx; no effects on cytotoxicity, potentially due to the activation of unfolded protein responses, as a consequence of triggered ER stress
Ghio et al., 2011 [80]	Human alveolar macrophages, human monocytes (THP-1 cell line), primary and immortalized (BEAS-2B cell line) human normal bronchial epithelial cells exposed to GdCl_3_ or Omniscan™ (gadodiamide)	A concentration-dependent uptake of Gd (III) was observed for all cell types, for both GdCl_3_ and gadodiamide; co-exposure of cells to GdCl_3_ and ferric ammonium citrate increased iron levels compared to incubation with each compound alone; in BEAS-2B cells, GdCl_3_ triggered increased production of IL-18, and co-exposure with ferric ammonium citrate led to increased ferritin levels
Long et al., 2011 [81]	Human adenocarcinoma cells (HeLa cell line) exposed to GdCl_3_	Cell proliferation and increased lipid and amino acid metabolisms at low concentrations, while promoting cell death and disrupting the metabolism of lipids, amino acids, and carbohydrates at higher concentrations
MacNeil et al., 2011 [82]	Primary human keratinocytes and dermal fibroblasts exposed to Gd-EDTA, Omniscan™ (gadodiamide), or Dotarem^®^ (gadoteric acid)	Gd-EDTA and gadodiamide stimulated both fibroblast and keratinocyte viability at lower concentrations and induced cell death at higher levels; they also stimulated collagen production in fibroblasts, but not in keratinocytes
Okada et al., 2011 [83]	Mouse pre-osteoblastic cells (MC3T3-E1 cell line), human adipose tissue-derived mesenchymal stem cells, human subcutaneous preadipocytes, and human dermal fibroblasts, exposed to GdCl_3_	Cell differentiation in all cell types and Ca^2+^ deposition, leading to abnormal calcification; downregulation of type I collagen was also observed in fibroblasts
Wang et al., 2011 [84]	Prostate cancer cells (DU145 and PC3 cell lines) exposed to GdCl_3_	Inhibition of PC3 cell viability via apoptosis (↑annexinV), as well as cell migration in both cell lines, which was mediated by the inactivation of both ERK1/2 and p38 MAPK pathways; increase in Ca^2+^ levels; all effects appear to be regulated upstream by the PTx-sensitive Gi protein signaling pathway; suppression of cell-induced osteoclast differentiation via the RANKL/RANK/OPG pathway
Wiesinger et al., 2011 [85]	Human umbilical vein endothelial cells (HUVECs) and human dermal fibroblasts (HSF 1 cells) were exposed to Gadovist^®^ (gadobutrol), Magnevist^®^ (gadopentetic acid), MultiHance^®^ (gadobenic acid), or Omniscan™ (gadodiamide), as well as the manganese- and the iron-based contrast agents Teslascan^®^ and Resovist^®^	Gadodiamide and Teslascan^®^ showed antiproliferative effect in HUVECs, which was rapidly compensated; HSF 1 cells showed no effect on TGFβ levels after exposure to the GBCAs
Xia et al., 2011 [86]	Primary cultured rat cortical neurons exposed to GdCl_3_	Cytotoxicity in neurons, with increased Ca^2+^ levels, through oxidative injury (↑ROS) and ER stress-related signal transduction
Bhagavathula et al., 2010 [87]	Human dermal fibroblasts and intact skin in organ culture exposed to GdCl_3_	Increased cell proliferation in fibroblasts, possibly involving MAPK/PI3K signaling pathways; upregulation of MMP-1 and TIMP-1 in both cells and skin culture; increased type 1 collagen deposition in the skin
Del Galdo et al., 2010 [88]	Human monocyte-derived macrophages incubated with Omniscan™ (gadodiamide)	Stimulated macrophage activation, with NFκB-dependent expression, and increased chemokines production (↑CCL2 and 8, CXCL10 and 11) and iNOS
Gou et al., 2010 [89]	Mouse macrophages (RAW 264.7 cell line) treated with GdCl_3_	No effect on macrophage viability; trigger of profibrotic/pro-inflammatory responses (↑TGFβ1 and IL-6) via the activation of protein kinase C and ERK1/2 signaling pathways
Li et al., 2010 [90]	Mouse embryo fibroblasts (NIH3T3 cell line) treated with Gd-containing particles	Promoted G_1_/S cell cycle progression through the activation of ERK and Akt signaling pathways; increased levels of serum in media led to the formation of smaller particles that exert a stronger effect on cell cycle
Feng et al., 2010 [91]	Primary cultures of embryonic cortical neurons exposed to GdCl_3_	Cell death by apoptosis (↓MT activity, ΔΨ_m_ and ATP, ↑cytochrome *c*, and caspase 3), oxidative stress (↑ROS), and DNA fragmentation
Bhagavathula et al., 2009 [92]	Human dermal fibroblasts treated with Omniscan™ (gadodiamide)	Increased production of MMP-1 and TIMP-1 and increased type I collagen deposition, without affecting type I procollagen production
Fu et al., 2009 [93]	Mouse embryo fibroblasts (NIH-3T3 cell line) exposed to GdCl_3_	Increased cell growth, promoting G_1_/S cell cycle progression (↑cyclin A, B, and D), which appears to be mediated by activation of both ERK and PI3K signaling pathways
Liao et al., 2009 [94]	Male Wistar rats treated with a single dose of GdCl_3_	Liver damage with disrupted carbohydrate metabolism (↓glycogen, ↑succinate, lactate, alanine, and betaine); no histological evidence of kidney damage, but with changes in renal metabolic profile
Moriconi et al., 2009 [95]	Male Wistar rats and C3H/HeJ endotoxin-resistant mice injected intraperitoneally with a single dose of GdCl_3_	Phagocytosis dysregulated the hepatic iron metabolism (↑hepcidin, ↓hemojuvelin, and ferroportin-1); these changes might be mediated by the locally produced acute-phase-cytokines (↑IL-1β and -6, TNFα)
Steger-Hartmann et al., 2009 [96]	Male Wistar rats treated either once, three, or eight times with a daily administration of Omniscan™ (gadodiamide)	A decrease in reticulocyte and an increase in monocyte counts; a decrease in albumin/globulin ratio; histological signs of renal damage and dermal fibrosis; Gd (III) was detectable in the skin, femur, and liver; trigger a pro-inflammatory response, which appears to increase vascular permeability (↑OPN, VEGF, CXCL2, CCL1 and 3, TNFα, and TIMP-1)
Varani et al., 2009 [97]	Human dermal fibroblasts and human skin in organ culture, isolated from adult volunteers, treated with Omniscan™ (gadodiamide), Magnevist^®^ (gadopentetic acid), MultiHance^®^ (gadobenic acid), or Prohance^®^ (gadoteridol)	GBCA exposure increased fibroblast proliferation, accompanied by increased production of MMP-1 and TIMP-1, but not of type I procollagen; similar effects were observed with gadodiamide exposure in ex vivo skin
Wermuth et al., 2009 [98]	Human peripheral blood monocytes incubated with Omniscan™ (gadodiamide), Gd-DTPA, or GdCl_3_	The three compounds stimulated a pro-inflammatory/profibrotic response (↑IL-4, 6, and 13, interferon γ, TGFβ, VEGF, αSMA, and type I collagen)
Heinrich et al., 2007 [99]	Pig kidney proximal tubular cells (LLC-PK1 cell line) incubated with Magnevist^®^ (gadopentetic acid), MultiHance^®^ (gadobenic acid), Dotarem^®^ (gadoteric acid), or Omniscan™ (gadodiamide)	All GBCAs induced concentration-dependent cell death; induction of necrosis and apoptosis was more evident for gadopentetic and gadobenic acid
Korolenko et al., 2006 [100]	Male CBA mice administered with a single dose of GdCl_3_	GdCl_3_ accumulated in liver macrophages lysosomes, leading to damage and a decrease in macrophage density
Liu et al., 2003 [101]	Mitochondria isolated from Laca mice liver and human normal liver cells (7701 cell line) exposed to lanthanides	Disruption of MT function (↑MT swelling and membrane fluidity, and ↓ΔΨ_m_); induction of apoptosis (↑cytochrome *c* release) with potential involvement of oxidative stress (↑ROS)
Greisberg et al., 2001 [102]	Cultured bovine chondrocytes, isolated from articular cartilage, exposed to Omniscan™ (gadodiamide)	Adverse changes in chondrocyte metabolism (↑matrix production, ↓cellular proliferation, ↑apoptosis)
Yongxing et al., 2000 [103]	Human peripheral blood lymphocytes, from a healthy male adult, exposed to Gd(NO_3_)_3_	DNA damage (↑micronuclei frequency, single stranded DNA breaks and unscheduled DNA synthesis)
Zhang et al., 2000 [104]	Single ventricular myocytes, isolated from hearts of male guineapigs, exposed to GdCl_3_	Non-voltage dependent inhibitory effect on both inward and outward ionic current, which appears to reflect gradual Gd (III) accumulation at the binding site of the Na^+^-Ca^2+^ exchanger protein that carries the current
Bales et al., 1999 [105]	Bovine adrenal chromaffin cells treated with Gd (III)	Enhancement of the Ca^2+^ -mediated catecholamine secretion by inhibiting Ca^2+^ efflux
Roman et al., 1999 [106]	Primary cultured rat hepatocytes and rat hepatoma cells (HTC cell line) exposed to Gd (III)	High inhibition of ATP release in liver cells, suggesting that Gd (III) might be an effective inhibitor of ATP-permeable channels
Adding et al., 1998 [107]	Male New Zealand white rabbits infused with GdCl_3_ for 25 min	Decrease in pulmonary vascular resistance, which appears to be partly due to inhibition of NO formation
Ferreira et al., 1998 [108]	Liver mitochondria isolated from male Sprague Dawley rats treated with a single dose of GdCl_3_	A reversible decrease in liver O_2_ consumption, accompanied by a decline in MT cytochromes *c*1 and *c*
Badger et al., 1997 [109]	Liver microsomes and hepatocytes isolated from control male and female Sprague Dawley rats and rats administered with a single dose of GdCl_3_	GdCl_3_ treatment reduced the activity of total hepatic microsomal cytochrome P450 and aniline hydroxylase; it also reduced the susceptibility of hepatocytes to the cytotoxicity induced by CCl_4_, but not by CdCl_2_
Spencer et al., 1997 [110]	Male and female Sprague Dawley rats treated with a single administration of GdCl_3_	Deposition in capillary beds of the lung and kidney, and in the liver and spleen, with signs of necrosis in both organs; phagocytosis by the mononuclear phagocytic system was also observed
Rai et al., 1996 [111]	Rats treated with a single dose of GdCl_3_	Distribution of Kupfer cells in the liver and changes in their phenotype towards a more pro-inflammatory one (↑TNFα, ↓IL-10)
Ruttinger et al., 1996 [112]	Male Sprague Dawley rats treated with a single dose of GdC1_3_	Lower phagocytic activity of Kupfer cells, which may be related to the increased inflammatory response (↑TNFα and IL-6)
Mizgerd et al., 1996 [113]	Rat alveolar macrophages exposed to GdCl_3_	Cell death by apoptosis
Laine et al., 1994 [114]	Rat atrial preparations, from male Sprague Dawley rats, incubated with GdCl_3_	Blocked voltage-gated calcium channels and inhibited stretch-activated atrial natriuretic peptide secretion
Mlinar and Enyeart 1993 [115]	Rat and human medullary thyroid carcinoma cells (6-23 (clone 6) and TT cell lines, respectively) exposed to trivalent metal cations	GdCl_3_ blocked the current through T-type voltage gated calcium channel by occlusion of the channel pore, and in a voltage-independent way

ACACA, acetyl-CoA carboxylase alpha; ADP, adenosine diphosphate; Akt, protein kinase B; Apaf-1, apoptotic peptidase activating factor 1; ATP, adenosine triphosphate; CCL, C-C motif chemokine ligand; CCR2, C-C chemokine receptor type 2; CPT1A, carnitine palmitoyltransferase 1A; CRP, C-reactive protein; CTGF, connective tissue growth factor; CXCL, chemokine (C-X-C motif) ligand 1; EDTA, ethylenediaminetetraacetic acid; EGFR, epidermal growth factor receptor; ER, endoplasmic reticulum; ERK, extracellular-signal-regulated kinase; FAK, focal adhesion kinase; FcεRI, Fc epsilon Receptor I; GADD153, growth arrest- and DNA damage-inducible gene 153; GBCAs, gadolinium-based contrast agents; GFR, glomerular filtration rate; GRP78, glucose-regulated protein 78; GSH, glutathione; GSSG, glutathione disulfide; IL, interleukin; iNOS, inducible nitric oxide synthase; LC3, microtubule-associated protein 1A/1B-light chain 3; LDH, lactate dehydrogenase; LPO, lipid peroxidation; MAPK, mitogen-activated protein kinases; MMP-1, matrix metalloproteinase-1; MT, mitochondrial; NFκB, nuclear factor kappa B; NOX4, NADPH oxidase 4; NPs, nanoparticles; NRF2, nuclear factor erythroid 2–related factor 2; OPG, osteoprotegerin; OPN, osteopontin; PI3K, phosphatidylinositol 3-kinase; RANK, receptor activator of nuclear factor kappa-Β; RANKL, receptor activator of nuclear factor kappa-Β ligand; ROS, reactive oxygen species; SOD, superoxide dismutase; TBARS, thiobarbituric acid reactive substances; TGFβ, transforming growth factor beta; TIMP-1, tissue inhibitor matrix metalloproteinase 1; TNFα, tumor necrosis factor alpha; VEGF, vascular endothelial growth factor; αSMA, smooth muscle alpha-actin; ΔΨ_m_, mitochondrial membrane potential.

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
