# Peer review of "Toxicity Mechanisms of Gadolinium and Gadolinium-Based Contrast Agents—A Review"

_ijms, 2024, doi:10.3390/ijms25074071_

Round 1

Reviewer 1 Report

Comments and Suggestions for Authors

The review entitled “Toxicity mechanism of gadolinium and gadolinium-based contrast agents-a review” is dealing with toxicity mechanisms of GBCAs. The review is well-structured and results presented are an important contribution to the scientific community interested both in the development of new GBCAs or in understanding safety of currently clinical approved GBCAs.

However, there are some points that could be improved before the publication:

1)      Authors should include in the review the following articles:

-Roberts DR, Lindhorst SM, Welsh CT, Maravilla KR, Herring MN, Adam Braun K, et al. High levels of gadolinium deposition in the skin of a patient with Normal renal function. Investig Radiol. 2016;51(1):280–289.

-Parant M, Sohm B, Flayac J, Perrat E, Chuburu F, Cadiou C, Rosin C, Cossu-Leguille C. Impact of gadolinium-based contrast agents on the growth of fish cells lines. Ecotoxicol Environ Saf. 2019 Oct 30;182:109385. doi: 10.1016/j.ecoenv.2019.109385. Epub 2019 Jun 28. PMID: 31260918.

-Ahmed Uosef, Arijita Subuddhi, Annie Lu, Henry V. Ubelaker, Christof Karmonik, Jarek Wosik, Rafik M. Ghobrial, Malgorzata Kloc, 7T MRI and molecular studies of Dotarem (gadoterate meglumine) retention in macrophages., Journal of Magnetic Resonance Open,Volumes 12–13, 2022, 100085, ISSN 2666-4410, https://doi.org/10.1016/j.jmro.2022.100085.

-Hall, A.J., Robertson, A.G., Hill, L.R. et al. Synthesis and tumour cell uptake studies of gadolinium(III)–phosphonium complexes. Sci Rep 11, 598 (2021). https://doi.org/10.1038/s41598-020-79893-9

-Di Gregorio E, Gianolio E, Stefania R, Barutello G, Digilio G, Aime S. On the fate of MRI Gd-based contrast agents in cells. Evidence for extensive degradation of linear complexes upon endosomal internalization. Anal Chem. 2013 Jun 18;85(12):5627-31. doi: 10.1021/ac400973q. Epub 2013 Jun 7. PMID: 23738707.

-Elham Mohammadi,  Massoud Amanlou,   Seyed Esmaeil Sadat Ebrahimi,   Morteza Pirali Hamedani,   Abdolkarim Mahrooz,  Bita Mehravi,  Baharak Abd Emami,   Mohammad Reza Aghasadeghi,  Ahmad Bitarafan-Rajabi,   Hamid Reza Pour Ali Akbarh   Mehdi Shafiee Ardestani. Cellular uptake, imaging and pathotoxicological studies of a novel Gd[iii]–DO3A-butrol nano-formulation. RSC Adv., 2014,4, 45984-45994

2)      In figure 1 the short name of the macrocyclic agents Gadobutrol needs to be corrected in Gd-DO3A.

3)      In the paragraph Gd(III) mechanism of toxicity, lines 82 to 87, for a better reading comprehension, please clarify more comprehensively how the studies have been divided.

4)      I strongly encourage writing few lines in the paragraph “concerns about the use of GBCA”, or creating a new paragraph, in which are outlined some solutions to reduce the toxicity given by the administration of GBCA such as the encapsulation in liposomes or nanoparticles. In this regard, please cite the following articles:

- Lamichhane N, Udayakumar T, D’Souza W, Simone Ii C, Raghavan S, Polf J, et al. Liposomes: clinical applications and potential for image-guided drug delivery. Molecules. 2018;23:288.

- Šimečková, P., Hubatka, F., Kotouček, J. et al. Gadolinium labelled nanoliposomes as the platform for MRI theranostics: in vitro safety study in liver cells and macrophages. Sci Rep 10, 4780 (2020). https://doi.org/10.1038/s41598-020-60284-z

- Asghar Narmani, Bagher Farhood, Hamed Haghi-Aminjan, Tohid Mortezazadeh, Akbar Aliasgharzadeh, Mehran Mohseni, Masoud Najafi, Hadis Abbasi, Gadolinium nanoparticles as diagnostic and therapeutic agents: Their delivery systems in magnetic resonance imaging and neutron capture therapy,Journal of Drug Delivery Science and Technology, Volume 44, 2018, Pages 457-466, ISSN 1773-2247, https://doi.org/10.1016/j.jddst.2018.01.011.

- Gallo E, Rosa E, Diaferia C, Rossi F, Tesauro D, Accardo A. Systematic overview of soft materials as a novel frontier for MRI contrast agents. RSC Adv. 2020 Jul 21;10(45):27064-27080. doi: 10.1039/d0ra03194a. PMID: 35515779; PMCID: PMC9055484.

- Mueller R, Moreau M, Yasmin-Karim S, Protti A, Tillement O, Berbeco R, Hesser J, Ngwa W. Imaging and Characterization of Sustained Gadolinium Nanoparticle Release from Next Generation Radiotherapy Biomaterial. Nanomaterials (Basel). 2020 Nov 13;10(11):2249. doi: 10.3390/nano10112249. PMID: 33202903; PMCID: PMC7697013.

Author Response

We thank the reviewer for the comments and suggestions, to which we reply point by point:

  1. The articles referred were included in the manuscript.
  2. Gadobutrol is also known as Gd-DO3A-butrol, which is why we have inserted this designation in the figure.
  3. Concerning data presented in Table 1, we rewrote the text about this issue, for a better understanding of the way they are presented.
  4. As suggested, at the end of “concerns about the use of GBCA”, we added some data concerning solutions to reduce the toxicity associated with exposure to GBCA, including the articles mentioned.

Reviewer 2 Report

Comments and Suggestions for Authors

The authors reviewed  the toxicity of GBCA typically used for magnetic resonance imaging.  They have conducted a literature survey relating the toxicity of GBCA for both invitro and in-vivo models  from 1993 to 2022; the main findings of reported results were listed in Table 1. They have addressed and discussed possible mechanisms arising from GBCA especially to the renal system. It appeared that linear linear GBCAs were more unstable and toxic comparing with macrocylic counterparts.  This  article also provided adequate references. So it can be accepted for publication in IJMS as it is. 

Author Response

We would like to thank the reviewer for his pleasant comments on our work.

Reviewer 3 Report

Comments and Suggestions for Authors

Dear Editor/Authors,

The manuscript ID: ijms-2924101 with the title: “Toxicity mechanisms of gadolinium and gadolinium-based contrast agents – a review” by the authors: Susana Coimbra, Susana Rocha, Nícia Reis Sousa, Cristina Catarino, Luís Belo, Elsa Bronze-da-Rocha, Maria João Valente and Alice Santos-Silva, present a study aimed to provide information about potential toxicity of gadolinium which is a component of Gadolinium-based contrast agents (GBCA) that is being used to improve magnetic resonance imaging. In this review paper, the authors considered the possible toxic effects of gadolinium, which has been used as an MRI agent for many years. the mechanisms of its deposition and in particular the influence on the function of the kidneys through which it is excreted are considered.

General comments:

Title

The title is appropriate, precise, and clear to readers.  

Abstract

Written clearly and understandably. In brief, includes all the elements for understanding MS

Introduction

In the introduction, the authors provide basic information about chelated gadolinium (III), its use and toxicity. The chelated form of Gd (III) is less toxic and retains its paramagnetic properties. The authors also list various forms of Gd (III) that can be used.They focus is on the deposition of Gd in different organs, such as kidney,  liver, brain, skin and bone tissue. Very high levels of Gd was found in kidney, heart and blood vessels. nephrogenic systemic fibrosis (NSF) is associated with different forms of Gd. Takođe su naveli da su apoptoza, oksidativni stres i upregulation of inflamacije glavni uzroci citotoksičnosti. The authors have presented a very interesting review of the side effects of Gd, which has been used in medicine for 30 years in MRI. Through a detailed database search, they have given us a chronological insight into 89 scientific papers on the toxicity of Gd, including both in vitro and in vivo studies. The authors identified signaling pathways involved in Gd toxicity. Due to the persistence of GBCA in tissue, the European Medicines Agency (EMA) has recommended restricting the use of GBCA. For me, the observation between lines 242-252 is particularly interesting. The authors note that the linear GBCAs are more unstable and therefore exhibit higher Gd (III) retention and cytotoxicity in the organs compared to GBCAs with a macrocyclic structure.

Considering the usefulness of contrast agents, the lack of safer alternatives to GBCA,

Conclusion of the Reviewer

It seems that, regardless of the side effects, especially in patients with damaged kidneys, the paradigm of "more good than harm" applies until an appropriate alternative is found.   I do not want my review to become an analysis or a eulogy.   Therefore, I have just one question for the authors: how many "the safe ones" MRIs can be done with GD (III)?   Even without this data, I think this review article is very important as it is a public health issue and I hope it encourages someone to look at appropriate alternatives to Gd (III).  

I suggest that the paper be accepted for publication in the present form.

Author Response

We would like to thank the reviewer for his pleasant comments on our work. The question raised by the reviewer is quite pertinent, “how many "the safe ones" MRIs can be done with GD (III)?“,

We adressed that subject in the final considerations of our manuscript:

”Although there are studies reporting nephrotoxicity and impaired renal function associated with repeated administrations of GBCA, the frequency of exposure used in most research studies poorly mimics the use of these agents in clinical practice, and some research studies were carried out in models of advanced stage of renal disease. Also, the use of different lengths of exposure to GBCA make difficult the interpretation and comparison between studies. The effect of repeated administrations in mild kidney disease using standardized exposures to contrast agents deserves further study.”

Actually, existing data are few, use different assay and clinical conditions and, thereby, are not conclusive. Further studies are indeed warranted, to unveil remaining doubts concerning the safety of repeated exposure to GBCA, whcih occurs often in clinical practice.

Reviewer 4 Report

Comments and Suggestions for Authors

The Authors did a great work to prepare this manuscript. However there are other papers on toxicity of GBCA and I suggest some changes to make this paper more valuable and interesting for thereaders.

It is better to divide the table into two seperate ones: one with results of in vivo studies, and second with the in vitro studies. It makes the results easier to analise.

I think it will be better to discuss the mechanism according to the toxic effects induces by gadolinium, for example fibrosis, neurotoxicity etc.

It will be better to add a Metodology section, in which search strategies should be described.

Author Response

We thank the reviewer for the comments and suggestions.

Some of the articles referred in table 1 include both in vitro and in vivo studies, the division of the table in two will duplicate information, for that reason we decided to maintain table 1 as it is. However, if the Reviewer and the Editor think that an alteration is relevant to the improvement of the article, we are willing to make it.

We did not create subheadings according to the organ or tissue affected, when discussing Gd (III) and GBCA mechanisms of toxicity, because the same type of mechanism was found for different types of cells/tissues. Nonetheless, we discussed data abording sequentially fibrosis, upregulation of inflammation, oxidative stress, apoptosis, interference with calcium homeostasis, as well as another mechanisms of toxicity.

At the first paragraph of “Gd (III) mechanisms of toxicity”, we described with more detail the methodology used to select the articles presented in our work. Since this a traditional review article, not a systematic or a scoping review, we thought this clarification would be more appropriate rather than creating a Methodology section.

Round 2

Reviewer 4 Report

Comments and Suggestions for Authors

The Authors answered to all my suggestion and made proper changes in the manuscript.

I have only one question. Line 69: What is the meaning of "aquatic environment"? The cited paper desribed effects on cell lines, not the whole organism, so it should be clarified here. Aquatic environment suggests that effect were studied in the environment similar to the natural one and on the organism.